# RILe: Reinforced Imitation Learning

## Abstract

Learning to imitate behaviors from a limited set of expert trajectories is a promising way to acquire a policy. In imitation learning (IL), an expert policy is trained directly from data in a computationally efficient way, but requires vast amounts of data. On the other hand, inverse reinforcement learning (IRL) deduces a reward function from expert data and then learns a policy with reinforcement learning via this reward function. Although this mitigates the data inefficacy problem of imitation learning, IRL approaches suffer from efficiency issues because of sequential learning of the reward function and the policy. In this paper, we combine the strengths of imitation learning and inverse reinforcement learning and introduce RILe: Reinforced Imitation Learning. Our novel dual-agent framework enables joint training of a teacher agent and a student agent. The teacher agent learns the reward function from expert data. It observes the student agent's behavior and provides it with a reward signal. At the same time the student agent learns a policy by using reward signals given by the teacher. Training the student and the teacher jointly in a single learning process offers scalability and efficiency while learning the reward function helps to alleviate data-sensitivity. Experimental comparisons in reinforcement learning benchmarks against imitation learning baselines highlight the superior performance offered by RILe particularly when the number of expert trajectories is limited.

## 1 Introduction

Learning to achieve human-level performance in complex tasks with artificial agents is a long-pursued goal in machine learning research. Reinforcement learning (RL) offers a solution to this problem by maximizing a utility/reward function through diverse interactions with the environment. However, this reward function must be meticulously tailored to the task to ensure that its maximization leads to optimal actions Sutton & Barto (2018). This becomes infeasible for complex tasks, where an agents needs to sequentially perform multiple subtasks. To bypass the need for reward engineering, one can learn task execution from expert demonstrations. Research proposed two primary approaches for this purpose: Imitation learning and Inverse Reinforcement Learning.

Imitation learning (IL) aims to learn a mapping from the current observation of the environment to action from given expert demonstrations. Imitation learning algorithms are developed to recover expert-like policies for tasks with large state-action spaces Hussein et al. (2017). While imitation learning approaches tackle the reward engineering limitation of RL, they struggle in generalizing beyond the provided expert trajectories. Thus, an extensive collection of high-quality expert demonstrations is essential for achieving good performance Zheng et al. (2022).

On the other hand, Inverse Reinforcement Learning (IRL) approaches aim to learn the intrinsic reward function of the expert. This reward function is used to guide an RL agent, enabling it to reproduce expert-like behavior. IRL generally suffers from scalability and inefficiency issues Zheng et al. (2022), since it relies on sequentially learning the reward function and the policy. These problems get exaggerated when the task gets complex, due to larger observation and action spaces.

In this work, we aim to bridge the gap between imitation learning and inverse reinforcement learning. We propose RILe, a novel approach for learning a reward function and a policy simultaneously. Our framework comprises two interacting agents: a student agent and a teacher agent. The teacher agent observes the student and provides a reward, which the student aims to maximize simultaneously. In return, the teacher is rewarded based on the similarity between the behavior of the student agent

and the behavior observed in a limited set of samples of expert trajectories. This setting enables the student to replicate expert behaviour without being trained on expert trajectories nor directly observing the similarity of its policy's behavior to that of an expert.

This architecture allows our framework to leverage the strengths of both IL and IRL, resulting in a hybrid approach that effectively compensates their respective limitations. Specifically, introducing the teacher agent as an intermediary between the policy learning and reward acquisition stages enables training the student agent in a standard RL setting while ensuring that its policies mimic expert behavior. This breaks the data-policy connection common to existing IL solutions Ho & Ermon (2016) and facilitates a less data-sensitive learning process that retains the generalization capabilities of standard RL, since RL agent does not try to overfit to data directly or via some similarity metric. Consequently, it can generalize over the specific state-action pairs of expert trajectories. In addition, the dual-agent setting enables simultaneous learning of the intrinsic reward function of the expert and the policy that replicates their behavior. This surpasses the limitation of iterative sequential learning of reward and policy common to IRL approaches. Our framework is capable of acquiring the reward function and policy in a single learning process.

To demonstrate efficacy, we compare our method to state-of-the-art of imitation learning and inverse reinforcement learning on two different benchmarks: Atari games (Bellemare et al., 2013) and MuJoCo control tasks (Todorov et al., 2012). Experimental results reveal that our approach outperforms baselines especially when the available expert data is limited. This indicates a better data-efficacy of our method compared to baselines.

## 2    RELATED WORK

We review the literature on learning expert behavior from demonstrations. Commonly, expert demonstrations are sourced either through direct queries to the expert in any observable state or by collecting sample trajectories demonstrated by the expert. We present related work that aligns with the most prevalent approaches of the latter setting, namely Imitation Learning and Inverse Reinforcement Learning. Both, IL and IRL, form the conceptual foundation of RILe.

Offline reinforcement learning also learns policies from data, which may include expert demonstrations. In contrast to our setting, its main goal is to learn a policy without any online interactions with the environment. We refer the reader to (Levine et al., 2020) for an overview of offline RL. Furthermore, hierarchical reinforcement learning (HRL) splits tasks into subtasks at different levels of temporal and functional abstraction (Sutton et al., 1999; Dayan & Hinton, 1992).While HRL has been combined with imitation learning Le et al. (2018), its goal is different to our setting as it abstracts long-horizon tasks to render them learnable instead of replicating expert behavior.

**Imitation Learning**   The earliest work on imitation learning introduced Behavioral cloning (BC) (Bain & Sammut, 1995), which aims to learn a policy congruent with expert demonstrations through supervised learning. SEARN introduces a classifier to BC that facilities exploring the observation space in continued training after cloning the expert policy (Daumé et al., 2009). DAgger proposes the aggregation of expert demonstrations with policy experiences during the training of the policy for improving generalization over expert demonstrations (Ross et al., 2011). Ho & Ermon (2016) introduced Generative Adversarial Imitation Learning (GAIL) where a discriminator aims to understand whether queried behavior stems from a policy or from expert demonstrations, while a generator tries to fool the discriminator by learning a policy that exhibits expert-like behavior. InfoGAIL extends upon GAIL by extracting latent factors from expert behavior and employing them during imitation learning (Li et al., 2017). Hester et al. (2018) proposed Deep Q-learning from Demonstrations (DQfD) where the learning agent is first pre-trained using expert demonstrations, followed by a subsequent policy optimization through interactions with the environment. Similarly, expert data is leveraged in Le et al. (2018); Kostrikov et al. (2019) to reduce number of environment interactions and increase learning efficacy. Zero-Shot Visual Imitation first learns a policy without considering expert demonstrations, and then uses expert data in a goal-conditioned setting to fine-tune the policy (Pathak et al., 2018). ValueDice proposes an off-policy imitation learning method using a distribution-matching objective between policy and expert behavior (Kostrikov et al., 2020).

Although the field of imitation learning has seen innovative advancements, the requirement for high-quality expert data and the need for data efficacy remain open challenges (Zheng et al., 2022).

Moreover, the limited generalization capability of IL approaches persists (Toyer et al., 2020). We address these limitations related to IL's data sensitivity by introducing an intermediary teacher agent, thereby breaking the direct connection between the policy and the expert demonstrations.

**Inverse Reinforcement Learning** In inverse reinforcement learning, Ng & Russell (2000) introduced three algorithms to learn the intrinsic reward function of an expert and acquire the expert policy from it. Apprenticeship learning builds on IRL and proposed to represent the reward function as a linear combination of features (Abbeel & Ng, 2004). Maxium Entropy Inverse Reinforcement Learning is proposed to deal with the noise in expert demonstrations and recover the expert reward function better (Ziebart et al., 2008). Several works extended IRL to include negative examples into the learning process In (Lee et al., 2016; Shiarlis et al., 2016; Bogert et al., 2016). Guided Cost Learning approximates the reward function with a neural network and makes maximum entropy methods applicable to continuous state-action spaces (Finn et al., 2016). An adversarial reward learning framework is proposed by Fu et al. (2018) to address the scalability issues of classical approaches. Chen et al. (2021) introduces a pipeline that makes IRL work with unstructured, real-world data. Cross-embodiment scenarios are considered in XIRL, opening up a new direction in IRL (Zakka et al., 2022).

Despite the advancements in IRL, the efficacy of the learning process and scalability to complex problems remain open challenges (Arora & Doshi, 2021). The main reason for these limitations is the iterative sequential learning framework employed in IRL. We solve this efficacy problem by learning the policy and reward function, via training a student agent and a teacher agent, in a single joint learning process.

## 3 BACKGROUND

### 3.1 PRELIMINARIES

Our work considers an imitation learning problem from expert trajectories. Each trajectory comprises states $s \in S$ and actions $a \in A$, where $S$ and $A$ are state and action spaces respectively. The set of expert trajectories is defined as $\tau_E = [[(s_0, a_0), (s1, a1), \ldots], [(s_0, a_0), (s1, a1), \ldots], \ldots]$, which are sampled from an expert policy $\pi_E \in \Pi$, where $\Pi$ is the set of all possible policies. $P(s'|s, a)$ is an unknown state transition probability function. The reward function $R(s, a)$ generates a reward given a state-action pair $(s, a)$. In this work, we consider $\gamma$-discounted infinite horizon settings. Following Ho & Ermon (2016), expectation with respect to the policy $\pi \in \Pi$ refers to the expectation when actions are sampled from $\pi(s)$: $E_\pi[R(s, a)] \triangleq E_\pi[\sigma_{t=0}^\infty \gamma^t R(s_t, a_t)]$, where $s_0$ is sampled from an initial state distribution $\rho(s)$, $a_t$ is given by $\pi(\cdot|s_t)$ and $s_{t+1}$ is determined by the transition model as $P(\cdot|s_t, a_t)$.

### 3.2 REINFORCEMENT LEARNING (RL)

Reinforcement learning seeks to find an optimal policy that maximizes the discounted cumulative reward. The reinforcement learning problem is defined as

$$RL(R_\theta) = \pi^* = \arg\max_\pi E_\pi[R_\theta(s, a)] = E_\pi[\sum_{t=0}^\infty \gamma^t R_\theta(s_t, a_t)], \tag{1}$$

where $R_\theta \in \mathbb{R}$ is a reward function parameterized by $\theta$ and the optimal policy is indicated by $\pi^*$. Regularization can be introduced with the entropy function $H(\pi)$. In this work $\gamma$-discounted casual entropy function is considered, which defined as $H(\pi) = E_\pi[-log\pi(a|s)]$ (Ho & Ermon, 2016; Bloem & Bambos, 2014). Incorporating entropy regularization into the problem transforms it into

$$RL(R_\theta) = \pi^* = \arg\max_\pi H(\pi) + E_\pi[R_\theta(s, a)]. \tag{2}$$

### 3.3 INVERSE REINFORCEMENT LEARNING (IRL)

Given sample trajectories $\tau_E$ of an expert policy $\pi_E$, inverse reinforcement learning, $IRL(\tau_E)$, tries to recover the reward function, $R^*$ that would result in expert behavior when optimized in a reinforcement learning training, $RL(R^*)$. In other words, the goal can be defined as

$$RL(R^*) = \pi^* = \arg\min_{\pi} E_{\tau_E}[L(\pi, \pi_E)] \tag{3}$$

where $L(\pi, \pi_E)$ is a loss function that measures difference between given policies. Inverse reinforcement learning seeks to find the reward function in which the expert policy performs better than any other policy.

$$IRL(\tau_E) = \arg\max_{R\in\mathbb{R}} \left( E_{\pi_E}[R(s,a)] - \max_{\pi} E_{\pi}[R(s,a)] \right) \tag{4}$$

With entropy regularization $H(\pi)$, maximum casual entropy inverse reinforcement learning (Ziebart et al., 2008) can be defined as

$$IRL(\tau_E) = \arg\max_{R\in\mathbb{R}} \left( E_{\pi_E}[R(s,a)] - \max_{\pi} \left( E_{\pi}[R(s,a)] + H(\pi) \right) \right) \tag{5}$$

### 3.4 ADVERSARIAL IMITATION LEARNING (AIL)

In contrast to inverse reinforcement learning, imitation learning aims to directly acquire the expert policy from given expert trajectory samples. It can be formulated as

$$IL(\tau_E) = \arg\min_{\pi} E_{\tau_E}[L(\pi(s), \pi_E(s))]. \tag{6}$$

GAIL (Ho & Ermon, 2016) extends imitation learning to an adversarial setting by quantifying the similarity between policies of the agent and the expert with a discriminator $D_\phi(s,a)$, parameterized by $\phi$. Its goal is to find the optimal policy that minimizes this difference metric while maximizing an entropy constraint by training the discriminator and the policy at the same time. The optimization problem can be formulated as a zero-sum game between the discriminator $D_\phi(s,a)$ and the policy $\pi$, represented by

$$\min_{\pi}\max_{\phi} E_{\pi}[log D_\phi(s,a)] + E_{\tau_E}[log(1 - D_\phi(s,a))] - \lambda H(\pi). \tag{7}$$

In other words, the reward function that is maximized by the policy is defined as a similarity function, expressed as $R(s,a) = -log(D_\phi(s,a))$.

### 3.5 PROBLEM FORMULATION

An Standard MDP is defined as $MDP_S : (S, A, R, T, K, \gamma)$ where $S$ is state-space, consist of all possible environment states, and $A$ is action space consists of all possible environment actions. $R = R(s,a) : SxA \rightarrow \mathbb{R}$ is the reward function. $T = \{P_{sa}\}$ is transition dynamics where $P_{sa}$ is defined as the state distribution upon taking action $a$ in state $s$. $K$ is initial state distribution, i.e. $s_0 \sim K$ and $\gamma$ is the discount factor. Another MDP is also defined, which can be stated as $MDP_T : (S_T, A_T, R_T, T_T, K, \gamma)$, where $S_T$ is state space defined as $SxA$, so consisting all possible state action pairs from $MDP_S$. $A_T$ is action space, a mapping from $S_T = (SxA) \rightarrow \mathbb{R}$, so the action is a scalar value. $R_T : S_T \rightarrow \mathbb{R}$ is only state-based reward. $T_T = \{P_{s_t a_t}\}$ is transition dynamics where $P_{s_t a_t}$ is defined as the state distribution upon taking action $a_t$ in state $s_t$. $K_T$ is initial state distribution, i.e. $s_{t,0} \sim K$. We assume that we have an access to $m$ expert trajectories, all of which have $n$ time-steps, $\zeta = \{s_0^{E,i}, s_1^{E,i}, \ldots, s_n^{E,i}\}_{i=1}^m$

# 4 RILe: Reinforced Imitation Learning

We propose Reinforced Imitation Learning (RILe) to combine the strengths of adversarial imitation learning and inverse reinforcement learning. The goal of the hierarchical framework is to learn the reward function of an expert and recover a policy that emulates expert-like behavior simultaneously in one learning process, without directly assessing the similarity between the behavior of the trained agent and the expert. Our framework consists of three key components: a discriminator, a student agent, and a teacher agent (Figure 1).

**Discriminator** The discriminator aims to understand whether a given state-action pair comes from an expert trajectory or not. It is defined as a feed-forward deep neural network, parameterized by $\phi$. Given expert state-action pairs $(s, a) \sim \zeta$ and other state-action pairs whose source is different than the expert data, $(s, a) \notin \zeta$, the discriminator aims to discriminate expert pairs from others. Thus, the optimization problem is defined as

$$\max_{\phi} E_{(s,a) \sim \zeta}[log(D_\phi(s,a))] + E_{(s,a) \notin \zeta}[log(1 - D_\phi(s,a))]. \tag{8}$$

**Student Agent** The student agent aims to learn a policy $\pi_S$ by interacting with an environment in a standard RL setting within $MDP_S$, where for each of its actions $a^S$ the environment returns a new state $s^E$. However, rather than from a hand-crafted reward function, the student agent receives its reward from the policy of the teacher agent $\pi_T$. Therefore, in $MDP_S$, the reward function is represented by the teacher policy $R = \pi_T$. The student agent is guided by the actions of the teacher agent, i.e., the action of the teacher is the reward of the student: $r^S = \pi_T((s^E, a^S))$. The optimization problem of the student agent is defined as

$$\min_{\pi_S} -E_{(s^E, a^S) \sim \pi_S}[\pi_T\left((s^E, a^S)\right)]. \tag{9}$$

The student agent aims to recover the optimal policy $\pi_S^*$ defined as

$$\pi_S^* = \arg\max_{\pi_S} E_{(s^E, a^S) \sim \pi_S}\left[\sum_{t=0}^{\infty} \gamma^t[\pi_T\left((s_t^E, a_t^S)\right)]\right]. \tag{10}$$

**Teacher Agent** The teacher agent aims to guide the student to mimic expert behavior by operating as its reward mechanism. Therefore, the teacher agent learns a policy $\pi_T$ that produces adequate reward signals to guide the student agent, by interacting with an environment in a standard RL setting within $MDP_T$. Since the state space of $MDP_T$ is defined over state-action pairs of $MDP_S$, the state of the teacher comprises the state-action pair of the student $s^T = (s^E, a^S)$. It generates a scalar action $a^T$ which is given to the student agent as reward $r^S$. The teacher agent's reward function, which depends only on its state, is defined as $R^T = Y$, where $Y$ is a reward approximating network. Therefore, the optimization problem of the teacher can be defined as

$$\min_{\pi_T} E_{s^T \sim \pi_S}[Y]. \tag{11}$$

The teacher agent aims to recover the optimal policy $\pi_T^*$ by maximizing the cumulative reward yielded through function $Y$:

$$\pi_T^* = \arg\max_{\pi_T} E_{(s^T) \sim \pi_S}\left[\sum_{t=0}^{\infty} \gamma^t[Y\left((s_t^T, )\right)]\right] = E_{(s^E, a^S) \sim \pi_S}\left[\sum_{t=0}^{\infty} \gamma^t[Y\left((s_t^E, a_t^S)\right)]\right]. \tag{12}$$

**RILe** RILe combines the three key components defined previously in order to converge to a student policy, which mimics expert behaviors presented in $\zeta$. To achieve this goal, the discriminator optimization problem is tweaked as

$$\max_{\phi} E_{(s,a)\sim\zeta}[log(D_{\phi}(s,a))] + E_{(s,a)\sim\pi_S}[log(1 - D_{\phi}(s,a))]. \tag{13}$$

In other words, the discriminator aims to discriminate state-action pairs from expert and student agent. This reformulated discriminator is employed as the reward function of the teacher $Y = log(D_{\phi})$, which translates the teacher's optimization problem to

$$\min_{\pi_T} E_{(s,a)\pi_S}[log(D_{\phi}(s,a))]. \tag{14}$$

In RILe, the student policy $\pi_S$ is trained with soft actor-critic (SAC) Haarnoja et al. (2018), with the aim of maximizing the cumulative rewards obtained from the teacher agent. Concurrently, the teacher agent $\pi_T$ is trained with proximal policy optimization (PPO) Schulman et al. (2017) to maximize the cumulative reward derived from the discriminator. Consequently, to increase its rewards, the teacher agent must encourage the student to generate state-action pairs that deceive the discriminator into perceiving them as originating from an expert. SAC is chosen to train the student policy to leverage past experiences and guidance from the teacher. PPO is utilized to train the teacher to enable fast adaptations to the changing feedback of the learning discriminator. The training algorithm is given in Appendix F.

To prove that the student agent can learn expert-like behavior, we need to show that the teacher agent learns to give higher rewards to student experiences that match with the expert state-action pair distribution, as this would enable a student policy to eventually mimic expert behavior.

**Lemma 1:** Given the discriminator $D_{\phi}$, the teacher agent optimizes its policy $\pi^{\theta_T}$ via policy gradients to provide rewards that guide the student agent to match expert's state-action distributions.

However, since the teacher is guided by a discriminator, we also need to show that the discriminator successfully learns to discriminate expert state-action pairs, i.e., understand whether the given state-action pair is generated by the expert or not.

**Lemma 2:** The discriminator $D_{\phi}$, parameterized by $\phi$ will converge to a function that estimates the probability of a state-action pair being generated by the expert policy, when trained on samples generated by both a student policy $\pi^{\theta_S}$ and an expert policy $\pi_E$.

The proofs of these lemmas are presented in Appendix 6.

## 4.1 INTUITION BEHIND RILE

In AIL, the learning agent, is guided by a discriminator that follows the definition presented in Eq. 13. However, in AIL, the student tries to satisfy the discriminator directly. Since the discriminator just aims to minimize a step-based cross entropy loss, it cannot consider the long-term effects of generated rewards. This myopic discriminator consequently leads to an agent that can mimic expert state-action pairs but cannot consider if its choices are optimal for long-horizon tasks. Moreover, such myopic strategies may also lead to failure in understanding connections between different possible states. In contrast, IRL incrementally updates the reward function and, at each iteration, re-trains a policy from scratch. Through this approach, IRL can learn the effect of reward signals on the behavior of a policy. However, iterative reward and policy training are inefficient, rendering IRL computationally infeasible for most real-world problems.

In RILe, we synergize advantages of IRL and AIL. Specifically, similar to IRL, we are learning the reward function via the teacher agent, and train a policy via the student agent that reflects updates in the reward function. However, we guide the teacher learning via an adversarial discriminator, inspired from AIL, and learn the reward function via RL to consider long-horizon effects of produced rewards. By introducing the adversarial discriminator, we can simultaneously learn a reward function and a policy simultaneously, rendering RILe computationally feasible, in contrast to IRL. Furthermore, the teacher agent learns to act based on long term effects of the produced reward signals and relations between different states by minimizing long-horizon costs within the the standard RL setting.

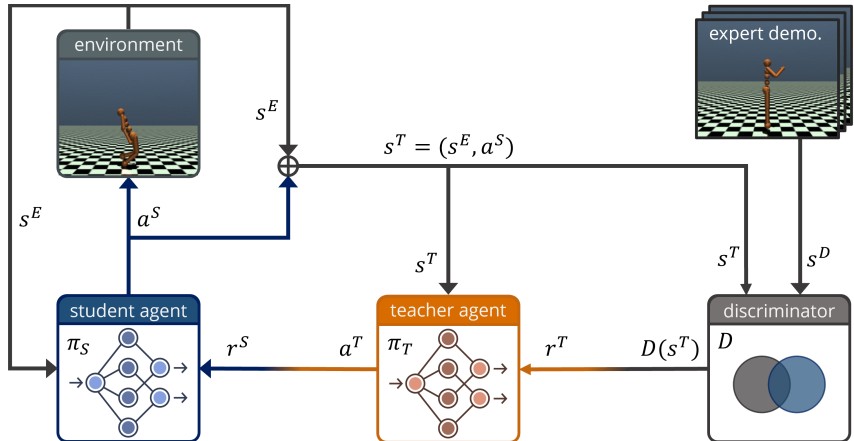

Figure 1: **Framework overview**. The framework consists of three key components: a student agent, a teacher agent, and a discriminator. The student agent learns a policy $\pi_S$ by interacting with an environment where for each of its actions $a^S$ the environment returns a new state $s^E$. It receives its reward from the teacher's policy $\pi_T$, which evaluates the state action pair of the student agent $s^T = (s^E, a^S)$ and chooses an action $a^T$ that then becomes the reward of the student agent $a^T = r^S$. The teacher agent is rewarded $r^T$ by a discriminator $D$ that tries to distinguish if a state stems from an agent ($s^T$) or from expert demonstrations ($s^D$). In a single learning process, our framework can learn policies that exhibit expert behavior without having direct access to expert demonstrations.

## 5 EXPERIMENTAL EVALUATION

### 5.1 EXPERIMENTAL SETUP

We evaluate RILe against baselines on different tasks from two different reinforcement learning benchmarks: (1) Atari games (Bellemare et al., 2013) and (2) MuJoCo control tasks (Todorov et al., 2012). For the MuJoCo benchmark, we purposely evaluated on control tasks of varying complexity, encompassing both low- and high-dimensional state spaces. For all experiments, OpenAI Gym is used as the simulation framework (Brockman et al., 2016). All the tasks are described in detail in supp. material.

To obtain expert trajectories, we utilize the experts from RL-Zoo3 (Raffin, 2020). Their policies were trained on the true cost function of each task, which are defined by Brockman et al. (2016). Different numbers of trajectories are sampled (e.g., 1 or 100 for Atari) from these trained experts to assess performance across a range of available expert demonstrations.

To ensure the visitation of states that are not present in expert demonstrations, all experiments are initialized randomly. This is further reinforced by the stochastic nature of the actions taken by the learning agents.

RILe is tested against an imitation learning-, an adversarial imitation learning- and an inverse reinforcement learning baseline, which are:

- Behavioral cloning (BC): Employed as the supervised imitation learning baseline.
- Generative Adversarial Imitation Learning (GAIL): GAIL is utilized as the adversarial imitation learning baseline.
- Adversarial Inverse Reinforcement Learning (AIRL): Utilized as the inverse reinforcement learning baseline.

For all baselines, we use their respective implementation of stable-baselines3 (Gleave et al., 2022).

Networks are randomly initialized at the start. In all tasks of both benchmarks, the policy of BC is trained for a 1000 epochs via supervised learning. All baselines are trained for 2 million time-steps, and RILe is trained for 1 million time-steps.

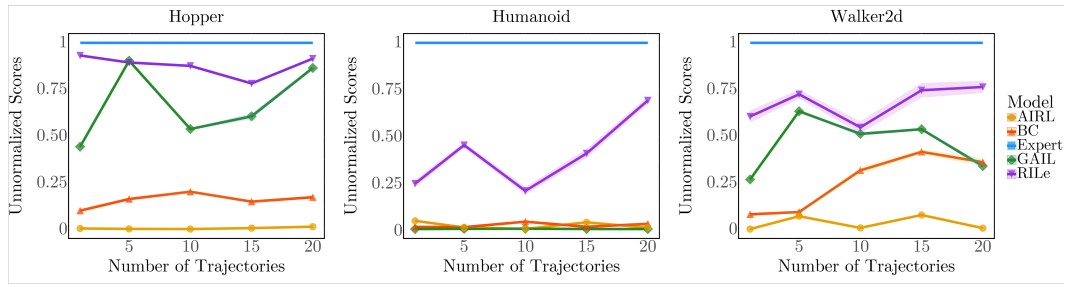

Figure 2: Mean ± std. error of the reward achieved on evaluation by RILe and baselines in MuJoCo control tasks.

## 5.2 ATARI

For the Atari benchmark, all methods are evaluated using two sets of expert demonstrations, comprising one expert trajectory and 100 expert trajectories, respectively. Instead of using an image-based observation state, all approaches use a vector representation of the RAM of the Atari emulator as state space. Moreover, frame-stacking is avoided and single frame observations are used, which changed the hardness of games.

In the case of discrete Atari tasks, we employ PPO (Schulman et al., 2017) as the learning agent for baselines and as the teacher and student agents in RILe. Hyperparameter sweeps and selected hyperparameters can be found in D The hyperparameters for PPO agents follow the default settings of stable-baselines3.

Table 1: Mean ± std. err. of the attained reward on test trajectories of RILe and baselines in Atari environments. Traj. stands for the number of available expert trajectories during training.

|  | Traj. | Asteroids | BeamRider | Qbert | SpaceInv. |
|---|---|---|---|---|---|
| RILe |  | **1960±33.4** | **458±18.7** | **270±34.8** | **279.8±6.5** |
| GAIL | 1 | 1402.8±19.8 | 330±34 | 125±3.4 | 222.1±10.9 |
| AIRL |  | 140±2.5 | 0±0 | 0±0 | 270±13.5 |
| BC |  | 1550± 20.3 | 264±6.8 | 150± 5.9 | 180±1.2 |
| RILe |  | **1904±45.9** | **498.4±15.8** | 315±17.3 | **295.59±12.7** |
| GAIL | 100 | 1729±38.5 | 409.2±22.3 | 125±2.6 | 235±3.1 |
| AIRL |  | 140±0.5 | 0±0 | 0±0 | 270±2.9 |
| BC |  | 1440±56.3 | 616±45.3 | **820.75±11.8** | 180±4.4 |

Table 1 presents the performance of RILe alongside the baselines. RILe performs better than baselines in all of the tasks as presented. An exception is Qbert, where behavioral cloning outperforms all other approaches when trained on 100 expert trajectories.

## 5.3 MUJOCO

In the case of the MuJoCo benchmark, methods are evaluated with five different sets of expert demonstrations: 1, 5, 10, 15, 20 expert trajectories. SAC (Haarnoja et al., 2018) is used as learning agent for baselines and for the student agent in RILe. RILe's teacher agent employs a PPO policy (Schulman et al., 2017). Hyperparameter sweeps and selected hyperparameters can be found in Appendix D.

The performance of RILe and the baselines in MuJoCo-based control tasks is presented in Figure 2. RILe outperforms baselines in all three tasks. This holds true even in the case of the Humanoid task, which involves a larger state-action space and greater complexity. The consistently superior performance of RILe across all three sets of expert demonstrations demonstrates that our method performs effectively even with a limited amount of data.

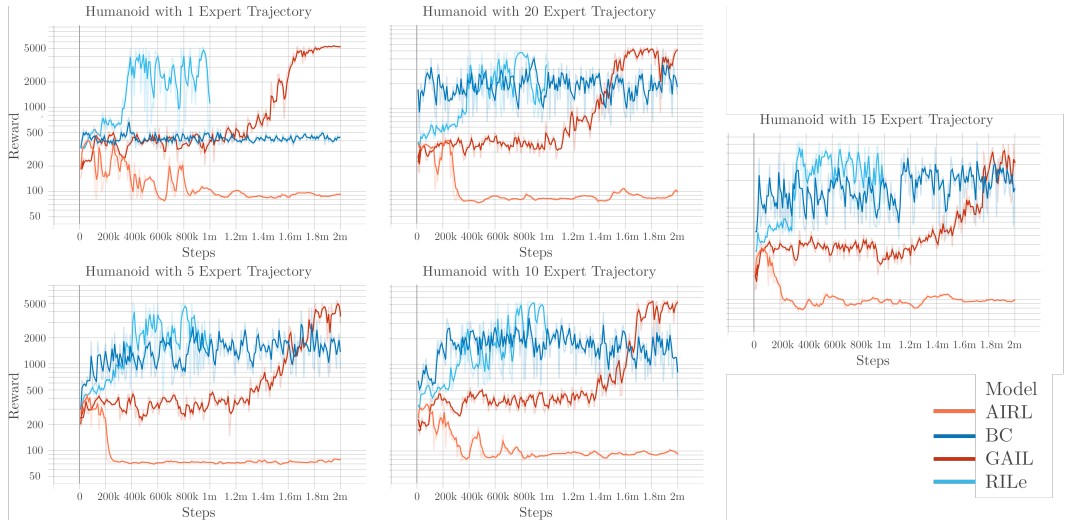

Figure 3: Mean rewards in repetitive evaluations during the training, that are basically obtained via allowing agents to take deterministic actions on their own training environments.

In order to compare the sample efficacy between methods, at the end of each 10000th time-step during training, methods are evaluated on their own environment, and results are presented in Figure 3 for Humanoid-v3 environment with different sizes of expert data. RILe is significantly more sample efficient when compared to AIRL and GAIL. Behavioral cloning also demonstrated superior sample efficacy, however, as presented in Figure 2, this doesn't translate into a better performance in tests, because of over-fitting.

## 6   DISCUSSION

We have demonstrated in the experiments that our method beats the baselines in different settings with different data availability and can perform well even with just one expert demonstration. This shows the data-efficacy of our method when compared to imitation learning and inverse reinforcement learning baselines. The experiments are conducted in ten different tasks, and all experiments are initialized randomly. RILe generalizes better than the baselines in states which are not included in the expert demonstrations. Since the policies of all approaches, including the student agent and the teacher agent are stochastic, the training eventually covers states which are not included in expert demonstrations, especially when the number of trajectories are small. Hence, the reported results indicate how robust policies are towards deviations from the expert demonstrations.

Although combining imitation learning and inverse reinforcement learning in RILE offers advantages, it also suffers from limitations. The main challenge is learning the reward function along with a policy, which means training the policy with a changing reward function. This inherently unstable setting can make the student agent get stuck in local minima, which results in sub-optimal behavior. To overcome this, we update the teacher agent less frequently by using a higher batch size compared to the student agent. Moreover, balancing the learning rates of the discriminator and the policies is difficult. For example, we have observed that for some training runs on the more challenging tasks of the MuJoCo benchmark, the teacher agent fails to satisfy the discriminator, since the latter converges exceptionally fast. This in turn makes it difficult for the teacher agent to find a reward for the student agent that tricks the discriminator. In such cases, the problem can be tackled by adjusting the learning rate of the discriminator or updating the discriminator less frequently. However, a more fundamental solution is required to optimally balance the different components of the architecture.

Future work should focus on improving the stability and unbalanced learning issues in RILe. One promising approach could consider using a learning curriculum or to learn an adaptable update frequency or learning rate for the discriminator.

## REPRODUCIBILITY STATEMENT

For the reproducibility of the results presented in this paper, all trained models along with scripts to generate results are provided as supplementary material. For the details of the experiments and used expert policies, refer to Appendix C. Detailed experimental results are presented in Appendix D.

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

# A  JUSTIFICATION OF RILE

## A.1  ASSUMPTIONS:

- The discriminator loss curve is complex and the discriminator function, $D_\phi(s, a)$, is sufficiently expressive since it is parameterized by a neural network with adequate capacity.
- For the teacher's and student's policy functions $(\pi^{\theta_T})$ and $(\pi^{\theta_S})$, and the Q-functions $(Q^{\theta_S})$, each is Lipschitz continuous with respect to its parameters with constants $(L_{\theta_T}), (L_{\theta_S}), and (L_Q)$, respectively. This means for all $(s, a)$ and for any pair of parameter settings $(\theta, \theta') : [|\pi^\theta(s,a) - \pi^{\theta'}(s,a)| \leq L_\theta|\theta - \theta'|,][|Q^\theta(s,a) - Q^{\theta'}(s,a)| \leq L_Q|\theta - \theta'|.]$

## A.2  PROOF FOR LEMMA 1

The expert's state-action distribution is denoted by $p_{\text{expert}}(s, a)$. The role of the teacher is to provide a reward signal to the student that encourages the approximation of $p_{\text{expert}}(s, a)$ as closely as possible.

We have $D_\phi : \mathcal{S} \times \mathcal{A} \to [0, 1]$ as the discriminator, parameterized by $\phi$, which outputs the likelihood that a given state-action pair $(s, a)$ originates from the expert, as opposed to the student. The teacher's policy $\pi^{\theta_T}$, parameterized by $\theta_T$, aims to maximize the likelihood under the discriminator's assessment, thus encouraging the student agent to generate state-action pairs drawn from a distribution resembling $p_{\text{expert}}(s, a)$.

The Value and Q functions of the teacher, conditioned on the rewards provided by the discriminator, are defined in terms of expected cumulative discriminator rewards. The value function for the teacher's policy parameters $\theta_T$ at state $s_t$ is given by:

$$V^{\theta_T}(s_t) = \mathbb{E}_{\pi^{\theta_T}} \left[ \sum_{k=0}^{\infty} \gamma^k r_{t+k} \mid s_t \right], \tag{15}$$

where $r_{t+k} = D_\phi(s_{t+k})$.

Similarly, the Q-function for taking action $a_t$ in state $s_t$ and then following policy $\pi^{\theta_T}$ can be written as:

$$Q^{\theta_T}(s_t, a_t) = D_\phi(s_t) + \gamma \mathbb{E}_{\pi^{\theta_T}} \left[ V^{\theta_T}(s_{t+1}) \mid s_t, a_t \right]. \tag{16}$$

The teacher's policy optimization is done by maximizing the following clipped surrogate objective function:

$$L^{CLIP}(\theta_T) = \mathbb{E}_{(s_t, a_t) \sim \pi^{\theta_T}} \left[ \min \left( \frac{\pi^{\theta_T}(a_t|s_t)}{\pi^{\theta_{T_{old}}}(a_t|s_t)} A^{\theta_{T_{old}}}(s_t, a_t), \right. \right.$$
$$\left. \left. \text{clip}(\frac{\pi^{\theta_T}(a_t|s_t)}{\pi^{\theta_{T_{old}}}(a_t|s_t)}, 1 - \epsilon, 1 + \epsilon) A^{\theta_{T_{old}}}(s_t, a_t) \right) \right], \tag{17}$$

where $A^{\theta_{T_{old}}}(s_t, a_t)$ is the advantage function computed as $Q^{\theta_{T_{old}}}(s_t, a_t) - V^{\theta_{T_{old}}}(s_t)$, defined with respect to the teacher's old policy parameters $\theta_{T_{old}}$.

By expressing the advantage using the reward from the discriminator, we explicitly tie the policy gradient updates to the discriminator's output, emphasizing the shaping of $\pi^{\theta_S}$ to match $p_{\text{expert}}(s, a)$:

$$A^{\theta_{T_{old}}}(s_t, a_t) = D_\phi(s_t, a_t) + \gamma \mathbb{E}\pi^{\theta_S} \left[ V^{\theta_{T_{old}}}(s_{t+1}) \right] - V^{\theta_{T_{old}}}(s_t). \tag{18}$$

During each policy update, the objective in Equation 17 is maximized, driving parameter updates to favor actions that elicit higher rewards from the discriminator – effectively the actions that better align with expert behavior:

$$\theta_T \leftarrow \theta_T + \alpha_\pi \nabla_{\theta_T} L^{CLIP}(\theta_T). \tag{19}$$

The update rule in Equation 19, driven by cumulative rewards from the discriminator, incrementally adapts the teacher's policy to reinforce student behaviors that are indistinguishable from those of the

expert according to $D_\phi$. The clipped surrogate objective and the trust-region policy optimization of PPO ensure that updates are conservative, preventing extreme shifts in the policy that could destabilize learning. Through these updates, the teacher's policy is mathematically guided to facilitate the student's approximation of $p_{\text{expert}}(s, a)$, thereby fulfilling its role in the imitation learning process.

### A.3    PROOF FOR LEMMA 2

The training objective for the discriminator is framed as a binary classification problem over mini-batches from expert demonstrations and student-generated trajectories. The discriminator's loss function $\mathcal{L}_D(\phi)$ is the binary cross-entropy loss, which for a mini-batch of size $n$ is defined as:

$$\mathcal{L}_D(\phi) = -\frac{1}{n} \sum_{i=1}^{n} \Big[ y_i \log(D_\phi(s_i, a_i)) + (1 - y_i) \log(1 - D_\phi(s_i, a_i)) \Big], \tag{20}$$

where $(s_i, a_i)$ are sampled state-action pairs from the combined replay buffer $\mathcal{D} = \mathcal{D}_S \cup \mathcal{D}_E$, with corresponding labels $y_i$ indicating whether the pair is from the expert ($y_i = 1$) or the student ($y_i = 0$). The stochastic gradient descent update rule for optimizing $\mathcal{L}_D(\phi)$ is then given by:

$$\phi \leftarrow \phi - \eta_D \nabla_\phi \mathcal{L}_D(\phi), \tag{21}$$

where $\eta_D$ is the learning rate for the discriminator.

Through iterative updates, $D_\phi(s, a)$ will converge to $P(\pi_E|s, a)$, provided the minimization of $\mathcal{L}_D(\phi)$ progresses according to the theoretical foundations of stochastic gradient descent. The convergence relies on the assumption that the student's policy $\pi^{\theta_S}$ and the expert policy $\pi_E$ induce stationary distributions of state-action pairs, such that the discriminator's data source is consistently representing both policies over time.

By minimizing $\mathcal{L}_D(\phi)$, we seek $\phi^*$ such that:

$$\phi^* = \arg\min_\phi \mathcal{L}_D(\phi). \tag{22}$$

Under typical conditions for convergence in stochastic gradient descent, the convergence to a local minimum or saddle point can be guaranteed. The discriminator's ability to distinguish between student and expert pairs improves as $\mathcal{L}_D(\phi)$ is minimized, implying that $\lim_{n_{batch} \to \infty} D\phi^*(s, a) = P(\pi_E|s, a)$, where $n_{batch}$ is the number of batches.

## B    TRAINING TRICKS

An overview of the training process is described in Algorithm 1. Training policies in a changing reward setting is challenging. However, we found that employing the following techniques is sufficient to train our three concurrently operating networks to convergence. To let the teacher agent learn well against a fast-learning discriminator, we maximize $E_{\pi_S}[log\,(D_\phi(s, a))]$ instead of minimizing $E_{\pi_S}[1 - log\,(D_\phi(s, a))]$ when training its policy, as original proposed by Goodfellow et al. (2014). To facilitate learning of the student agent, we reduce the update frequency of the teacher agent by utilizing a larger batch size compared to the student agent. This allows the student agent to gather experiences over a longer period of time with a consistent, non-updated reward function. In other words, the student agent is more-frequently trained when compared to reward agent, to have a more stable training for the student, and to let the teacher understand results of it's own actions better.

## C    TASKS AND EXPERTS

For the experimental tasks, Atari Bellemare et al. (2013) games and MuJoCo Todorov et al. (2012) based control tasks from OpenAI Gym Brockman et al. (2016) are used. The version numbers, observation and action space dimensions are presented in table 2. In Atari games, instead of images, 128 Bytes of RAM of the emulator is used as states, which can be considered as the extracted features. No frame-skipping is utilized, which means stochastically skipping some number of frames during the game by repeating the last action continuously.

Table 2: Tasks

|  | Task | Observation Space | Action Space |
|---|---|---|---|
| Atari | Asteroids-ramNoFrameskip-v4 | 128 (continuous) | 14 (discrete) |
|  | BeamRider-ramNoFrameskip-v4 | 128 (continuous) | 9 (discrete) |
|  | Breakout-ramNoFrameskip-v4 | 128 (continuous) | 4 (discrete) |
|  | Pong-ramNoFrameskip-v4 | 128 (continuous) | 6 (discrete) |
|  | Qbert-ramNoFrameskip-v4 | 128 (continuous) | 6 (discrete) |
|  | SpaceInvaders-ramNoFrameskip-v4 | 128 (continuous) | 6 (discrete) |
| MuJoCo | Hopper-v3 | 11 (continuous) | 3 (continuous) |
|  | Humanoid-v3 | 376 (continuous) | 17 (continuous) |
|  | Walker2d-v3 | 17 (continuous) | 6 (continuous) |

As the expert agents, RL-Zoo3 (Raffin, 2020) experts are utilized which are trained with stable-baselines3 policies on the true cost function of tasks, defined by OpenAI Gym Brockman et al. (2016). For Atari tasks, since the action space is discrete, PPO experts are utilized while for control tasks with continuous state-action spaces, SAC agents are used. The performance of expert policies are presented in Table 3. For Atari games, sessions are finished after 10000 time-steps, even the final state is not terminal. For control tasks, the maximum session length is defined as 1000 by Brockman et al. (2016), therefore sessions are naturally terminated after 1000 time-steps. Normalizations of state-vectors are done in control tasks for baselines, since it affected the performance drastically in a positive way.

Table 3: Expert scores and standard errors in experiment tasks

| Task | Mean Score | Std. Error | Mean Length | Std. Error |
|---|---|---|---|---|
| Asteroids-ramNoFrameskip-v4 | 1902.8 | 134.5 | 795.9 | 41.7 |
| BeamRider-ramNoFrameskip-v4 | 5001.9 | 331.6 | 4222.4 | 199.4 |
| Breakout-ramNoFrameskip-v4 | 73.2 | 27.5 | 9383.8 | 417.9 |
| Pong-ramNoFrameskip-v4 | 21.0 | 0 | 1655.8 | 1.77 |
| Qbert-ramNoFrameskip-v4 | 17739 | 328.1 | 2019.7 | 63 |
| SpaceInvaders-ramNoFrameskip-v4 | 1213.4 | 26.2 | 1461.4 | 57.6 |
| Hopper-v3 | 3598.9 | 211.4 | 522.8 | 59.7 |
| Humanoid-v3 | 6251.3 | 6 | 1000 | 0 |
| Walker2d-v3 | 3876.1 | 13.9 | 1000 | 0 |

## D    EXTENDED RESULTS

For all experiments, extended scores are provided in this section. Results for Atari environments are presented in Table 4, and results for control tasks are shown in Table 5, 6, and 7. For Atari tasks, sessions are finished after 10000 time-steps, even the final state is not terminal. For control tasks, the maximum time-steps is defined as 1000 as OpenAI Gym Brockman et al. (2016).

Note that GAIL performed better than RILe in both Hopper and Walker tasks when TRPO is being utilized as the learning agent. However, this is related to the inductive bias of GAIL, which is explained in Kostrikov et al. (2018). For instance, GAIL perform better than %20 the expert in Hopper and Walker with only one trajectory, which is unreasonable and most probably the result of the inductive bias.

Table 4: Experiment results for Atari environments

| Environment | Model | Num. Traj. | Mean Score | Std. Score | Mean Length | Std. Len. |
|---|---|---|---|---|---|---|
| Asteroids | RILe | 1 | 1960 | 33.4 | 4651.9 | 116.4 |
| | | 100 | 1904 | 19.8 | 5523.7 | 203.5 |
| | GAIL | 1 | 1402.8 | 45.9 | 3837.2 | 83.4 |
| | | 100 | 1729 | 38.5 | 3948.6 | 64.1 |
| | AIRL | 1 | 140 | 2.5 | 4144 | 5.5 |
| | | 100 | 140 | 0.5 | 4144 | 2.2 |
| | BC | 1 | 1550 | 20.3 | 3026 | 58.1 |
| | | 100 | 1440 | 56.3 | 4144 | 110.6 |
| BeamRider | RILe | 1 | 458 | 18.7 | 5173 | 45.6 |
| | | 100 | 498.4 | 15.8 | 5861 | 89.6 |
| | GAIL | 1 | 330 | 34 | 6147.2 | 69.0 |
| | | 100 | 409.2 | 22.3 | 6276.0 | 56.3 |
| | AIRL | 1 | 0 | 0 | 10000 | 0 |
| | | 100 | 0 | 0 | 10000 | 0 |
| | BC | 1 | 264 | 6.8 | 3612.8 | 102.3 |
| | | 100 | 616 | 45.3 | 9138 | 342.1 |
| Qbert | RILe | 1 | 270 | 34.8 | 1457.7 | 23.5 |
| | | 100 | 315 | 17.3 | 1680.3 | 27.3 |
| | GAIL | 1 | 125 | 3.4 | 1456 | 10.2 |
| | | 100 | 125 | 2.6 | 1456 | 7.4 |
| | AIRL | 1 | 0 | 0 | 1160 | 0 |
| | | 100 | 0 | 0 | 1160 | 0 |
| | BC | 1 | 150 | 5.9 | 1319.7 | 30.2 |
| | | 100 | 820.6 | 11.8 | 3746 | 105.3 |
| SpaceInvaders | RILe | 1 | 279.8 | 6.5 | 2667.4 | 53.3 |
| | | 100 | 295.6 | 12.7 | 2891.8 | 96.8 |
| | GAIL | 1 | 221.1 | 10.9 | 2165.2 | 71.4 |
| | | 100 | 235 | 3.1 | 2402.1 | 34.6 |
| | AIRL | 1 | 270 | 13.5 | 2151 | 85.2 |
| | | 100 | 270 | 2.9 | 2151 | 29.8 |
| | BC | 1 | 180 | 1.2 | 1565 | 5.5 |
| | | 100 | 180 | 4.4 | 1565 | 10.7 |

Table 5: Experiment results for Hopper-v3

| Environment | Model | Number of Trajectories | Mean Score | Std. Error Score | Mean Length | Std. Error Length |
|---|---|---|---|---|---|---|
| Hopper | RILe | 1 | 3354.8 | 17.18 | 1000 | 0 |
| | | 5 | 3219.2 | 13.1 | 1000 | 0 |
| | | 10 | 3154.5 | 3.7 | 1000 | 0 |
| | | 15 | 2815.9 | 10.4 | 1000 | 0 |
| | | 20 | 3294.84 | 11.34 | 1000 | 0 |
| | GAIL-SAC | 1 | 1597.7 | 27.4 | 928 | 14.9 |
| | | 5 | 3253.92 | 6.91 | 1000 | 0 |
| | | 10 | 1938.6 | 14.2 | 585.5 | 4.2 |
| | | 15 | 2183.1 | 57.8 | 679.5 | 17.4 |
| | | 20 | 3115.2 | 38.9 | 959.3 | 12.2 |
| | GAIL-TRPO | 1 | 3620.9 | 4.7 | 1000 | 0 |
| | | 5 | 3672.8 | 3.1 | 1000 | 0 |
| | | 10 | 3630.2 | 3.7 | 1000 | 0 |
| | | 15 | 3594.4 | 17 | 1000 | 0 |
| | | 20 | 12.2 | 0.1 | 14.1 | 0.1 |
| | BC | 1 | 366.4 | 12.1 | 138.3 | 3.6 |
| | | 5 | 590.2 | 76.1 | 198 | 2.8 |
| | | 10 | 731.1 | 25.5 | 230.1 | 7 |
| | | 15 | 540.6 | 17.5 | 181.6 | 4.5 |
| | | 20 | 625.1 | 61.8 | 200.8 | 16.1 |
| | AIRL-SAC | 1 | 23 | 0.5 | 21.1 | 0.3 |
| | | 5 | 15.42 | 0.5 | 18.2 | 0.4 |
| | | 10 | 12.5 | 0.4 | 14.2 | 0.4 |
| | | 15 | 29.8 | 1 | 25.9 | 0.5 |
| | | 20 | 47.4 | 4.9 | 39.8 | 2.1 |
| | AIRL-TRPO | 1 | 3.67 | 0.3 | 6.6 | 0.5 |
| | | 5 | 3.4 | 0 | 6 | 0 |
| | | 10 | 3.8 | 0.3 | 6.3 | 0.4 |
| | | 15 | 4.2 | 0.5 | 7 | 0 |
| | | 20 | 4.3 | 0.4 | 6.6 | 0.5 |

Table 6: Experiment results for Humanoid-v3

| Environment | Model | Number of Trajectories | Mean Score | Std. Error Score | Mean Length | Std. Error Length |
|---|---|---|---|---|---|---|
| Humanoid | RILe | 1 | 1575.4 | 127.8 | 307.1 | 24.7 |
| | | 5 | 2853.9 | 92.3 | 585.6 | 18.9 |
| | | 10 | 1334.3 | 12.4 | 263.9 | 24.4 |
| | | 15 | 2578.3 | 14.9 | 520.3 | 30.3 |
| | | 20 | 4338.1 | 12.9 | 848.7 | 25.3 |
| | GAIL-SAC | 1 | 66.3 | 0.1 | 14 | 0 |
| | | 5 | 73.1 | 5.1 | 15.1 | 0.7 |
| | | 10 | 77.6 | 2.4 | 16.4 | 0.5 |
| | | 15 | 68.3 | 2.3 | 14.4 | 0.5 |
| | | 20 | 66.3 | 0.1 | 14 | 0 |
| | GAIL-TRPO | 1 | 116.5 | 4.9 | 25.4 | 0.8 |
| | | 5 | 546.6 | 14.3 | 115.6 | 3 |
| | | 10 | 103.6 | 4.6 | 20.4 | 0.9 |
| | | 15 | 209.3 | 5.5 | 41.2 | 1.1 |
| | | 20 | 198.2 | 6.4 | 3.6 | 0.4 |
| | BC | 1 | 138.3 | 4.9 | 25.4 | 0.8 |
| | | 5 | 129.1 | 1.7 | 26.3 | 0.4 |
| | | 10 | 311.6 | 19.7 | 59.7 | 3.6 |
| | | 15 | 141.1 | 6.2 | 26.2 | 0.9 |
| | | 20 | 239.1 | 23.6 | 24.3 | 0.4 |
| | AIRL-SAC | 1 | 338.3 | 38.2 | 67.6 | 8.4 |
| | | 5 | 120.8 | 7 | 26.3 | 3.2 |
| | | 10 | 71.5 | 0.1 | 15.0 | 0 |
| | | 15 | 290.2 | 10.2 | 54.9 | 2.1 |
| | | 20 | 112.2 | 2.3 | 24.3 | 0.5 |
| | AIRL-TRPO | 1 | 68.1 | 2.3 | 14.4 | 0.5 |
| | | 5 | 75.2 | 1.5 | 15.9 | 0.3 |
| | | 10 | 144.9 | 3.8 | 28.2 | 0.7 |
| | | 15 | 149.4 | 4.2 | 34.9 | 2.1 |
| | | 20 | 141.6 | 7 | 30.6 | 1.6 |

Table 7: Experiment results for Walker2d-v3

| Environment | Model | Number of Trajectories | Mean Score | Std. Error Score | Mean Length | Std. Error Length |
|---|---|---|---|---|---|---|
| Walker2d | RILe | 1 | 2348.7 | 54.5 | 1000 | 0 |
| | | 5 | 2808.4 | 27.2 | 957.6 | 8.8 |
| | | 10 | 2892.1 | 25.3 | 930 | 18.9 |
| | | 15 | 2892.1 | 25.5 | 976.8 | 6.9 |
| | | 20 | 2960.2 | 8.5 | 1000 | 0 |
| | GAIL-SAC | 1 | 1040.8 | 28.9 | 998.9 | 3 |
| | | 5 | 2446.5 | 68.7 | 1000 | 0 |
| | | 10 | 1986.5 | 43.6 | 929.7 | 21.1 |
| | | 15 | 2083.4 | 15.3 | 972.3 | 6.8 |
| | | 20 | 1319.1 | 59.7 | 640.7 | 26.6 |
| | GAIL-TRPO | 1 | 4254.3 | 28.9 | 998.9 | 3.3 |
| | | 5 | 3048.8 | 70.9 | 841.5 | 19.9 |
| | | 10 | 3372.5 | 99.4 | 825.5 | 23.1 |
| | | 15 | 2743.3 | 83.6 | 747.9 | 21.1 |
| | | 20 | 2947.1 | 8.3 | 1000 | 0 |
| | BC | 1 | 317.9 | 85.3 | 140.5 | 29.2 |
| | | 5 | 365.4 | 23.5 | 165.2 | 6.9 |
| | | 10 | 1230.4 | 77.1 | 368.4 | 18.8 |
| | | 15 | 1614.1 | 61 | 441.1 | 13.5 |
| | | 20 | 349.2 | 12.2 | 149 | 3.2 |
| | AIRL-SAC | 1 | 12.6 | 0.5 | 22.4 | 0.5 |
| | | 5 | 278.9 | 6.7 | 162.2 | 6.9 |
| | | 10 | 45.7 | 0.8 | 43.4 | 0.5 |
| | | 15 | 304.9 | 9.6 | 181.9 | 9.4 |
| | | 20 | 31.6 | 0.9 | 40.6 | 0.5 |
| | AIRL-TRPO | 1 | -2.8 | 0.1 | 7 | 0 |
| | | 5 | -5 | 0.1 | 7 | 0 |
| | | 10 | -2.5 | 0.2 | 10 | 2.3 |
| | | 15 | -4.6 | 0.2 | 8 | 3.5 |
| | | 20 | -4 | 0.1 | 6 | 1.1 |

# E HYPERPARAMETERS AND HYPERPARAMETER SWEEPS FOR MUJOCO

Table 8: Hyperparameter Sweep and Best Hyperparameters for RILe in MuJoCo experiments

| Hyperparameters | Value |
| --- | --- |
| Number of discriminator updates per round | 2, **8** |
| The number of samples in each batch of expert data | 128 |
| Generator replay buffer size for discriminator updates | 100000 |
| Normalization | False |
| Discriminator Network Architecture | [256FC, 256FC] |
| Discriminator Optimizer | Adam |
| SAC Buffer Size | 1e5 |
| SAC Batch Size | 32 **256** |
| SAC Network Architecture | [256FC, 256FC] |
| SAC Activator Function | ReLU |
| SAC Discount Factor ($\gamma$) | 0.99 |
| SAC Learning Rate | 0.0003 |
| SAC Tau ($\tau$) | 0.005 |
| SAC Targer Entropy | 0.2 |

Table 9: Hyperparameter Sweep and Best Hyperparameters for GAIL in MuJoCo Experiments

| Hyperparameters | Value |
| --- | --- |
| Number of discriminator updates per round | **2**, 8 |
| The number of samples in each batch of expert data | **128**, 256 |
| Generator replay buffer size for discriminator updates | 64, **4096**, 8192 |
| Normalization | False, **True** |
| Discriminator Network Architecture | [32FC, 32FC] |
| Discriminator Optimizer | Adam |
| SAC Buffer Size | **1e5**, 1e6 |
| SAC Batch Size | 32, **256** |
| SAC Network Architecture | [256FC, 256FC] |
| SAC Activator Function | ReLU |
| SAC Discount Factor ($\gamma$) | 0.99 |
| SAC Learning Rate | 0.0003 |
| SAC Tau ($\tau$) | 0.005 |
| SAC Targer Entropy | Automatic |
| TRPO Number of steps per update | 1000 |
| TRPO Batch Size | 256 |
| TRPO Network Architecture | [256FC, 256FC] |
| TRPO Discount Factor ($\gamma$) | 0.99 |
| TRPO Learning Rate | 0.0001 |
| TRPO Lambda ($\lambda$) | 0.95 |
| TRPO Number of critic updates per policy update | 10 |
| TRPO Target Kullback-Leibler divergence | 0.01 |

Table 10: Hyperparameter Sweep and Best Hyperparameters for AIRL in MuJoCo experiments

| Hyperparameters | Value |
| --- | --- |
| Number of discriminator updates per round | 2, **8**, 16 |
| The number of samples in each batch of expert data | **128**, 256, 512 |
| Generator replay buffer size for discriminator updates | 64, **4096**, 8192, 16384 |
| Normalization | False, **True** |
| Discriminator Network Architecture | [32FC, 32FC] |
| Discriminator Optimizer | Adam |
| SAC Buffer Size | **1e5**, 1e6 |
| SAC Batch Size | 32, **256** |
| SAC Network Architecture | [256FC, 256FC] |
| SAC Activator Function | ReLU |
| SAC Discount Factor ($\gamma$) | 0.99 |
| SAC Learning Rate | 0.0003 |
| SAC Tau ($\tau$) | 0.005 |
| SAC Targer Entropy | Automatic |
| PPO Number of steps per update | 1000 |
| PPO Batch Size | **64**, 256, 8192, 16384 |
| PPO Network Architecture | **[256FC, 256FC]**, [64FC, 64FC] |
| PPO Discount Factor ($\gamma$) | 0.99 |
| PPO Learning Rate | 0.0003 |
| PPO Lambda ($\lambda$) | 0.95 |
| PPO Number of critic updates per policy update | 10 |
| PPO Entropy coefficient | 0.0 |
| PPO Value function coefficient | 0.5 |
| PPO Clipping parameter | 0.2 |

# F ALGORITHM

---

**Algorithm 1** RILe Training Process

---

1: Initialize student policy $\pi_S$ and teacher policy $\pi_T$ with random weights, and the discriminator $D$ with random weights.
2: Initialize an empty replay buffer $B$
3: **for** each iteration **do**
4:     Sample trajectory $\tau_S$ using current student policy $\pi_S$
5:     Store $\tau_S$ in replay buffer $B$
6:     **for** each transition $(s, a)$ in $\tau_S$ **do**
7:         Calculate student reward $R^S$ using teacher policy:

$$R^S \leftarrow \pi_T \tag{23}$$

8:         Update $\pi_S$ using policy gradient with reward $R^S$
9:     **end for**
10:     Sample a batch of transitions from $B$
11:     Train discriminator $D$ to classify student and expert transitions

$$\max_D E_{\pi_S}[\log(D(s, a))] + E_{\pi_E}[\log(1 - D(s, a))] \tag{24}$$

12:     **for** each transition $(s, a)$ in $\tau_S$ **do**
13:         Calculate teacher reward $R^T$ using discriminator:

$$R^T \leftarrow \log(D(s, a)) \tag{25}$$

14:         Update $\pi_T$ using policy gradient with reward $R^T$
15:     **end for**
16: **end for**

---

