# OpenReview forum: "RILe: Reinforced Imitation Learning"
_ICLR.cc/2024/Conference — Submitted to ICLR 2024_

### Official Review · Reviewer_3koB · 2023-10-26

**Soundness:** 2 fair
**Presentation:** 3 good
**Contribution:** 2 fair
**Rating:** 3
**Confidence:** 5

**Summary:**

This paper proposes RILe, which is a teacher-student imitation learning architecture by introducing an intermediary teacher agent into the GAIL optimization.

**Strengths:**

1. The framework seems interesting and new
2. The algorithm is simple and seems to be better than previous baselines, but I have questions about those results, see below.

**Weaknesses:**

1. The motivation and advantage of this work compared with previous methods is not clear.
2. Results are simply not good enough. For example, on Atari, most of the results do not solve the task (like -20 in Pong actually makes no difference with -21) and is similar to BC (or even worse); on Mujoco, the results are not consistent with those reported by the previous works.
3. Writing is clear the most of time.

**Questions:**

1. What is the motivation of such a teacher-student framework? By adding the complexity to GAIL, what is the advantage compared with it? Under Eq (12) the authors said
``This enables us to train the student agent in a standard RL setting where it receives rewards from the teacher to ensure that its policies mimic expert behavior. Thus, we can break the data-policy connection common to existing IL solutions and facilitate a less data-sensitive learning process that can generalize over the specific state-action pairs of expert trajectories.`` I do not see any `break the data-policy connection common` to GAIL, and I also do not understand why such a learning style `facilitate a less data-sensitive learning process`. GAIL also has such advantages.

2. `This reward structure allows us to utilize any single-agent reinforcement learning algorithm, instead of using supervised learning to optimize over loss functions defined in Equations 11 and 12.` What is the difference to GAIL? GAIL also allows any single-agent reinforcement learning algorithm. And why should we conduct `supervised learning over loss functions defined in Equations 11 and 12`? What are you specifying?

3. In Table 1, the Atari experiments
    - (a) Why the BC behaves better for traj. num. = 1 than traj. num. = 100? Especially on Asteroids and Qbert.
    - (b) Why the std for all baselines is 0? How many training/testing seeds are you using? How do you calculate the averaged results?

4. In Figure 2, the Mujoco experiments
    - (a) What is 2.5 expert trajectories?
    - (b) Why the results of baselines, like GAIL and AIRL is not consistent with the report from their paper or many previous imitation learning papers? What implementation are you using? What kind of expert demo are you using? From my experience, GAIL can easily solve Hopper and Walker.
    - (c) How many training/testing seeds are you using?

---

> ### Author Response · Authors · 2023-11-23
>
> We are grateful for the reviewer’s insightful comments and are convinced that they have improved our paper. Below, we address each of the concerns:
> 1. Motivation and Advantages of the Teacher-Student Framework:
> • We thank the reviewer for the comment. GAIL does indeed have advantages over behavioral cloning, since it tries to give rewards to the learning agent in states which are not covered in data through a discriminator (which is trained in a supervised manner). However, long-term effects of given rewards are never considered by the discriminator, since it is trained in a supervised way to minimize a step-based loss. This results in a discriminator that does not consider long term effects of the reward signals it produces, and eventually results in an agent which just tries to exactly imitate expert state-action pairs, even if they are problematic. For instance, in the dataset, for some specific state-action pairs, if all samples in data have a great noise on them, then the discriminator expects the student to exactly match these pairs, which results in a poor policy. Moreover, a discriminator may not fully understand connections between different possible states, since it does not maximize a cumulative loss, and since the loss just depends on the current step. In RILe, we consider long-horizon effects of given rewards by training a teacher agent as a reward function, which is rewarded based on a trajectory of a student. This also allows the teacher to learn long term effects of the produced reward signals and relations between different states, since the long-horizon cost is minimized. By breaking the data-policy connection, we imply that the student does not just mimics expert behavior, but satisfies an intermediary agent that learns that even if some state-action pairs are not beneficial in the short-term, these pairs may offer long-term benefits. Furthermore, instead of training a discriminator, which just considers the current timestep similarity of the given state-action pair, we train a the teacher agent by rewarding it on the trajectory generated by the student. All in all, the alleviation of the data sensitivity comes from long-horizon optimization problem of the reward, and the trajectory based loss used in teacher training. We have updated the method section to more clearly explain this motivation.
> 2. Reward Structure and Use of Reinforcement Learning Algorithms:
> • We removed the ¡supervised learning over loss functions defined in Equations 11 and 12¿ sentence to avoid confusion. We were trying implying that any single-agent RL method can be used to learn the reward function, instead of learning it via a supervised discriminator.
> 3. Concerns About Atari Experiments:
> • Note that we utilized Ram observations without frame stacking. This resulted in signif- icantly poorer performance than previously reported results, since the dynamics of the environment are hardly understandable without temporal information. In case of Pong and BreakOut, the lack of temporal data led to policies generating scores that imply a possible lack of convergence. Therefore, we excluded the experiments involving those games from the paper.
> • The random sampling of batches might cause BC to require more time to converge with 100 trajectories compared to a single one.
> • There was a misinterpretation in reporting the standard deviation for baselines, which we have now rectified. Updated results with correct standard deviations are included. We utilized ten different testing seeds for both RILe and baseline methods. Five training trials with varied random seeds were conducted for each setup, selecting the most promising outcome for testing.

---

> > ### Author Response · Authors · 2023-11-23
> >
> > 4. Concerns and Clarification About MuJoCo Experiments:
> > • The notion of 2.5 expert trajectories has been removed from the revised manuscript to avoid confusion. Instead, we have added three new, straightforward data sizes, offering a clearer representation of the training volume.
> > • We used RL Baselines3 Zoo experts, which are trained using stable-baselines3 implemen- tations and with optimized hyperparameters. There are different reasons behind the poor performance of baselines, especially for GAIL. We trained all methods, including ours, for 1 million time-steps, to have a sample-efficacy comparison to some extent. Moreover, in their original papers, GAIL and AIRL use TRPO, as learning agents. However, in this work, all baselines use SAC as the learning agent. In order to have a better and clear comparison, we included baseline results for extended trainings (with 2 million time-steps) in MuJoCo tasks. Moreover, results with their original learning agents, i.e., TRPO, are also included in Appendix D with a discussion on GAIL’s inductive bias in MuJoCo environments we are using. Another point behind the poor performance is the discrepancy between expert agents. We don’t have an access to expert agents used in GAIL paper, but their experts are significantly better than stable-baselines3 experts, which also effects the performance. The expert scores are also presented along with methods to have a relative comparison. More- over, the number of trajectories we are using in the experiments were small when compared to the original paper.
> > • 10 different test seeds are used. For the baselines and for our method, trainings are repeated 5 times with random seeds (which are selected randomly such that the seed number is larger than 100 or lower than 10), and the best one is reported.

---

### Official Review · Reviewer_RQBA · 2023-10-28

**Soundness:** 2 fair
**Presentation:** 3 good
**Contribution:** 2 fair
**Rating:** 5
**Confidence:** 4

**Summary:**

This paper proposes an interesting method to the imitation learning problem. It uses existing Adversarial Imitation Learning method to train a teach policy, and uses the action probability of the teach policy as a reward to train the (behavioral) student policy. Experiments are carried out on several Mujoco and Atari environments and outperform previous AIL/IRL methods like GAIL and AIRL.

**Strengths:**

1. The proposed idea is simple and straightforward.

2. The paper is easy to follow, and strong empirical performance is achieved on the Mujoco benchmark.

**Weaknesses:**

1. The major weakness of this paper is that is in its soundness. The teacher policy is trained to maximize the distribution matching between the student policy's state action visiting distribution and the expert's distribution, and its output probability is used as the reward for the student policy. It is unclear whether the student policy can still perform correct distribution matching. I would like to see some theoretical results on this. For the current version the method looks more like a heuristic but not a theoretically sound approach.

2. It is unclear to me why adding a teacher policy as a reward processor to train the behavior policy would be better than using AIL/IRL reward directly. Besides theoretical results, I would also like to see some intuitive & in-depth discussions on this point. Also, it looks like the choice of AIL/IRL reward for training the teacher policy should not be limited to GAIL, but the authors only base their method on GAIL. The authors can try to use different choices to see if their proposed approach can lead to some improvement.

3. I am also wondering the online sample efficiency of the proposed method. I think it would be necessary to add a return plot for comparison.

**Questions:**

See the weakness parts.

---

> ### Author Response · Authors · 2023-11-23
>
> We would like to express our gratitude for the insightful and constructive feedback provided on our paper. We have revised the paper to address each of the points raised and outline the updates we have made in response to the reviewers’ feedback below:
>
> 1. Presentation and Soundness:
> • We have thoroughly investigated the soundness of our approach. We have updated the Section 4 for better explanation of RILe. Each component is explained in more detail, by also providing their own optimization problems. Then, RILe is explained as the combi- nation of the three presented components: the student agent, the teacher agent and the discriminator. Moreover, we demonstrated the ability of the teacher to guide student for mimicking expert behaviors through a lemma, and provided theoretical results for this in Appendix A. This lemma shows that the student policy should eventually perform correct distribution matching to satisfy the teacher. We also have introduce another lemma that presents that the discriminator successfully learns to discriminate expert state-action pairs, and we provided theoretical results for this in Appendix A.
>
> 2. Comparison to AIL/IRL:
> • In AIL, the learning agent is guided by a discriminator, which is defined the same way in RILe, however, in our case it is used to guide the teacher agent instead of the learning agent. Therefore, in AIL, the student tries to satisfy the discriminator directly. Since the discriminator just aims to minimize a step-based cross entropy loss, the long-term effects of generated rewards are never considered by the discriminator. This results in a myopic discriminator which does not consider long term effects of the reward signals in the policy, and eventually results in an agent which is guided to exactly imitate expert state- action pairs, even if they are noisy. For clarification, in both AIL and RILe, the student agent is long-horizoned, maximizing a cumulative reward. What we are comparing is the discriminator in GAIL and the teacher agent in RILe. The myopic optimization in AIL may also lead to failure in understanding connections between different possible states. In other words, since the discriminator in RILe is just step-based, it does not have any incentive to discover the relation between states, i.e., it does not know what the next-state of the student is given the current state-action pair. However, the teacher agent in RILe is trained via RL, which helps it to discover these state-relations along with helping to understand long-term effects of produced reward signals in the end policy. In contrast, IRL considers the effects of the deduced reward function in a more exten- sive manner, by incrementally updating the reward function and re-training a policy from scratch. This sequential re-training of the policy helps IRL to fully understand the effect of reward signals on the policy. However, this incremental update pipeline comes with a bur- den of computational in-feasibility. Because for each small update in the reward function, a new policy is trained from scratch to see the effects of the reward fully. In RILe, we combined the granular understanding of the effects of the reward function representation from IRL, along with the adversarial discriminator from AIL. The aim of RILe is making IRL computationally feasible, by guiding the reward function learning with an adversarial discriminator. This results in a reward function, i.e., teacher, that learns long term effects of the produced reward signals and relation between different states, since long- horizon cost is minimized via RL. Thanks to guidance from the adversarial discriminator, the reward function is learned at the same time with the student policy. In Section 4, we have added a subsection Intuition, where we discuss the intuition behind RILe. In this section, we put RILe into context with adversarial imitation learning and inverse reinforcement learning intuitively.
>
> 3. Online Sample Efficiency:
> • Return plots for the Humanoid task are included in the updated paper under Section 5, providing a clear comparison of the online sample efficiency between RILe and baselines. We selected Humanoid since it is more challenging and requires more samples for training. In the plots, we present that when the same learning agent is used, which is SAC, RILe has significantly better sample efficiency.

---

### Official Review · Reviewer_KLBr · 2023-10-28

**Soundness:** 1 poor
**Presentation:** 1 poor
**Contribution:** 2 fair
**Rating:** 3
**Confidence:** 5

**Summary:**

This paper proposes an approach to imitation learning that combines approaches from inverse reinforcement learning and more standard imiation learning methods. The approach involves training a discriminator to distinguish between a teacher's actions and the expert actions, then using the teacher to "distill" information into a student model by using the teacher's action as a reward function. The authors evaluate their method on Atari and MuJoCo tasks, where they show that their method outperforms GAIL, AIRL, and BC.

**Strengths:**

- I believe that the approach of combining GAN-like training of a teacher with distillation into a student model is novel.
- The experimental results show that RILe has a substantial improvement over prior baselines on Atari and MuJoCo tasks.

**Weaknesses:**

- The presentation of this paper needs significant improvement, especially in the technical sections. It uses ambiguous and incorrect notation at times, and I am unsure what exactly the reward for the student is exactly. Several specific examples include:

- Abstract: "expert policy is trained directly from data in an efficient way, but requires vast amounts of data." -> This seems like a contradictory statement. Requiring vast amounts of data is normally not seen as "efficient".
- Sec. 3: Pi* is overloaded between equation 1 and equation 3 and refer to the optimum for different objectives.
- The notation IRL(t) should be defined before equation 3.
- Sec 4: Thus, it evaluates the state action pair of the student agent s T = (sE, aS) and chooses an action aT that, in turn, becomes the reward of the student agent aT = rS. -> This is a syntax error - rewards are scalar values, which an action is generally not.
- Eq 8 -> Same issue. What does it mean to minimize an action? Do you mean to minimize MSE between the action of the student and the expert?
- "the action of the teacher is the reward of thestudent: rS = πT ((sE, aS)" -> missing a closing parentheses.

- The performance of the baselines looks extremely poor (worse than performance on the same tasks presented in the original papers).This seems indicative of poorly tuned baselines. I do not completely trust the results until the questions I have raised in the questions section have been addressed.

**Questions:**

Sec 5.3: What does it mean to use 2.5 expert trajectories? What does half a trajectory mean?

Experiments: What was the performance (total reward) of the expert trajectories used for each experiment? This would be a good number to show as an upper bound / oracle of performance.

The section mentions different amounts of expert trajectories used, but only one table is reported. Are all of the different # trajectories averaged into the same table?

Please report learning rates, other hyperparameters, and hyperparameter sweeping strategies in the experiments or appendix (esp for the different components: the student, teacher, and discriminator).

Why do the reported baselines (AIRL, GAIL) perform so poorly, even poorer than performance reported in the original papers? For example, GAIL reports expert performance on the Walker and Hopper tasks in their paper, yet in the baselines reported here the performance is very poor. Would appreciate if the authors shed some insight into the discrepancy in results (e.g. is it because of the reduced number of expert trajectories?)

---

> ### Author Response · Authors · 2023-11-23
>
> We thank the reviewer for taking the time to provide a detailed review of our paper. We greatly appreciate the insightful comments and have made several improvements to the manuscript. Please find below our responses to the issues raised:
> 1. Presentation and Notation Clarity:
> • We have thoroughly revised Section 4 to clarify the technical description of our method. First, we explained three components (the student agent, the teacher agent and the dis- criminator) along with their own optimization problems for clarity.
> The discriminator aims to understand whether a given state-action pair comes from an expert trajectory or not. It is defined as a feed-forward deep neural network. Given expert state-action pairs and other state-action pairs whose source is different than the expert data, the discriminator aims to discriminate expert pairs from others.
> The student agent aims to learn a policy by interacting with an environment in a standard RL setting, where for each of its actions the environment returns a new state. However, rather than from a hand-crafted reward function, the student agent receives its reward from the policy of the teacher agent. Therefore, the reward function is represented by the teacher policy. The student agent is guided by the actions of the teacher agent, i.e., the action of the teacher is the reward of the student.
> The teacher agent aims to guide the student to mimic expert behavior by operating as its reward mechanism. Therefore, the teacher agent learns a policy that produces adequate reward signals to guide the student agent, by interacting with an environment in a standard RL setting. The state of the teacher comprises the state-action pair of the student. It generates a scalar action which is given to the student agent as reward. The teacher agent’s reward function is defined with a reward approximating network
> RILe combines the three key components defined previously in order to converge to a student policy, which mimics expert behaviors. To achieve this goal, the discriminator optimization problem is tweaked as to allow it to discriminate expert state-action pairs when samples from student’s and expert’s trajectories are given. In other words, the discriminator aims to discriminate state-action pairs from expert and student agent. This reformulated dis- criminator is employed as the reward function of the teacher. Consequently, to increase its rewards, the teacher agent must encourage the student to generate state-action pairs that deceive the discriminator into perceiving them as originating from an expert.
> To prove that the student agent can learn expert-like behavior, we need to show that the teacher agent learns to give higher rewards to student experiences that match with the expert state-action pair distribution, as this would enable a student policy to eventually mimic expert behavior.
> Lemma 1: Given the discriminator, the teacher agent optimizes its policy via policy gra- dients to provide rewards that guide the student agent to match expert’s state-action dis- tributions.
> Lemma 2: The discriminator will converge to a function that estimates the probability of a state-action pair being generated by the expert policy, when trained on samples generated by both a student policy and an expert policy.
> Theoretical results for lemmas are provided in Appendix A.

---

> > ### Author Response · Authors · 2023-11-23
> >
> > 2. Baseline Performance Discrepancies:
> > • Clarification on Implementation: For the baselines, we have used the Imitation library, which is built upon and referred by stable-baselines3. Our expert demonstrations are generated by RLbaselines3Zoo experts.
> > • Reasons behind the poor performance: There are several reasons behind the poor perfor- mance of baselines in MuJoCo, especially for GAIL. First, baselines and our method were initially trained for only 1 million time-steps to enable a comparison of sample efficiency. We recognize this limitation and have now included results from extended trainings of 2 million time-steps in MuJoCo tasks to address concerns about the baselines’ performance. Moreover, unlike GAIL and AIRL, which originally used TRPO, we had employed SAC for all methods to maintain consistency. We now present additional results in Appendix D using the original agents’ TRPO, to offer a clear and fair comparison. Lastly, the size of the dataset used in experiments were relatively small compared to baselines’ papers. It is one of the main reasons behind the relatively poor performance in Humanoid, since in their paper GAIL is trained with at least 80 trajectories. Although GAIL failed to work well with small number of trajectories, RILe converged to a well-performing policy. Moreover, in the inital submission, the data consisted of imperfect trajectories (trajectories whose total reward is less than %75 of the mean expert score) since it is expected from methods to learn well even if the trajectories are imperfect. This may also cause baselines to perform poorly. Therefore, these imperfect trajectories are removed, and the quality of the data is improved.
> > Note that, in Atari games, we utilized Ram observations without frame stacking. This resulted in significantly poorer performance than previously reported results, since the dy- namics of the environment are hardly understandable without temporal information. In case of Pong and BreakOut, the lack of temporal data led to policies generating scores that imply a possible lack of convergence. Therefore, we excluded the experiments involving those games from the paper.
> >
> > 3. Clarification on Expert Trajectories:
> > • We have removed references to fractional trajectories to avoid any confusion. The stated ”2.5 expert trajectories” have been excluded from the revised manuscript. To clarify, this setting encompassed two complete trajectories and half of another trajectory of the expert.
> >
> > 4. Expert Performance:
> > • We have incorporated the total reward performance of the expert trajectories utilized in each experiment as an upper bound to the figures and tables.
> >
> > 5. Representation of Results from Multiple Trajectories:
> > • Results from varying numbers of expert trajectories for MuJoCo tasks are now presented in the Appendix D. The increased number of varying trajectories allowed us to present that RILe is more robust to varying data sizes, keeping a stable performance. For Atari tasks, results from two different numbers of expert trajectories are reported in the main table.
> >
> > 6. Reporting of Hyperparameters:
> > • We have added all hyperparameters for RILe and the baselines to the Appendix E along with hyperparameter sweeps.

---

### Meta-Review · Area_Chair_pYpQ · 2023-12-06

**Metareview:**

This paper seeks to realize the benefits of direct policy imitation learning and inverse reinforcement learning using teacher and student agents. The reviewers appreciated the novelty of this dual agent formulation and the experimental validation of the approach. However, there were many criticisms about the clarity of the motivation/approach and the overall soundness. Though the author(s) made substantial revisions to address many of these concerns, more is needed to convince the reviewers for acceptance.

**Justification For Why Not Higher Score:**

The reviewers had many concerns about technical clarity, motivation, and soundness that prevent recommending acceptance.

**Justification For Why Not Lower Score:**

N/A

---

### Decision · Program_Chairs · 2024-01-16

Reject